# Evaluation of dental pulp stem cells response to flowable nano-hybrid dental composites: A comparative analysis

Dina Rady[1,2], Nassreen Albar[3], Waad Khayat[4], Mennatullah Khalil[5,6], Shereen Raafat[7,8], Mohamed Ramadan[9], Shehabeldin Saber[8,10,11] *, Mohamed Shamel[12]

1 Faculty of Dentistry, Oral Biology Department, Cairo University, Cairo, Egypt, 2 Faculty of Dentistry, Stem Cells and Tissue Engineering Research Group, Cairo University, Cairo, Egypt, 3 Restorative Department/ Operative, College of Dentistry, Jazan University, Jazan, Saudi Arabia, 4 Department of Restorative Dentistry, College of Dentistry, Umm Al-Qura University, Mecca, Saudi Arabia, 5 Hamdan Bin Mohamed College of Dental Medicine, Mohammed Bin Rashid University of Medicine and Health Sciences, Dubai Healthcare City, Dubai, United Arab Emirates, 6 Faculty of Dentistry, Dental Biomaterials Department, Fayoum University, Fayoum, Egypt, 7 Faculty of Dentistry, Pharmacology Department, The British University in Egypt (BUE), El Sherouk City, Egypt, 8 Dental Science Research Group, Health Research Centre of Excellence, The British University in Egypt (BUE), El Sherouk City, Egypt, 9 Specialized Dental Hospital, Armed Forces Medical Complex, Cairo, Egypt, 10 Faculty of Dentistry, Department of Endodontics, The British University in Egypt (BUE), El Sherouk City, Egypt, 11 Faculty of Dentistry, Department of Endodontics, Ain Shams University, Cairo, Egypt, 12 Faculty of Dentistry, Oral Biology Department, The British University in Egypt, El Sherouk City, Egypt

* shehabeldin.saber@bue.edu.eg

## Abstract

### Background

Flowable resin composites (FRC) are tooth-colored restorative materials that contain a lower filler particle content, and lower viscosity than their bulk counterparts, making them useful for specific clinical applications. Yet, their chemical makeup may impact the cellular population of the tooth pulp. This in-vitro study assessed the cytocompatibility and odontogenic differentiation capacity of dental pulp stem cells (DPSCs) in response to two recent FRC material extracts.

### Methods

Extracts of the FRC Aura easyflow (AEF) and Polofil NHT Flow (PNF) were applied to DPSCs isolated from extracted human teeth. Cell viability of DPSCs was assessed using MTT assay on days 1, 3 and 7. Cell migration was assessed using the wound healing assay. DPSCs' capacity for osteo/odontogenic differentiation was assessed by measuring the degree of mineralization by Alizarin Red S staining, alkaline phosphatase enzyme (ALP) activity, and monitoring the expression of osteoprotegerin (*OPG*), RUNX Family Transcription Factor 2 (*RUNX2*), and the odontogenic marker dentin sialophosphoprotein (*DSPP*) by RT-PCR. Monomer release from the FRC was also assessed by High-performance liquid chromatography analysis (HPLC).

**Data Availability Statement:** All relevant data are within the manuscript and its Supporting Information files.

**Funding:** The author(s) received no specific funding for this work.

**Competing interests:** The authors have declared that no competing interests exist.

**Abbreviations:** AEF, Aura easyflow; ALP, Alkaline phosphatase; ALPB, alkaline phosphatase buffer; ANOVA, Analysis of variance; ARS, Alizarin Red-S; Bis-GMA, bisphenol A-glycidyl methacrylate; cDNA, Complementary deoxyribonucleic acid; D3MA, decanediol dimethacrylate; DMEM, Dulbecco's Modified Eagle Medium; DMSO, dimethyl sulfoxide; DPSCs, Dental pulp stem cells; DSPP, odontogenic marker dentin sialophosphoprotein; FACS, Facial Action Coding System; FBS, Fetal bovine serum; FITC, fluorescein isothiocyanate; FRC, Flowable resin composites; HEMA, hydroxyethyl methacrylate; HPLC, high-performance liquid chromatography; IRB, institutional review board; LC, liquid chromatography; LED, light-emitting diode; mABs, labeled monoclonal antibodies; mRNA, messenger ribonucleic acid; MSCs, mesenchymal stem cells; MTT, 3-[4,5-dimethylthiazol-2-yl]-2,5 diphenyl tetrazolium bromide; OM, Induction/osteogenic medium; OPG, Osteoprotegerin; PBS, Phosphate Buffered saline; PE, phycoerythrin; PNF, Polofil NHT Flow; p-NP, para-nitrophenolate; p-NPP, para-nitrophenolate phosphate; RT-qPCR, Real-time Quantitative Reverse-transcriptase Polymerase Chain Reaction; RUNX2, RUNX Family Transcription Factor 2; SPSS, Statistical Package for the Social Sciences; TEGDMA, Trimethylene glycol dimethacrylate; UDMA, Urethane dimethacrylate; wt%, Percentage by weight.

## Results

DPSCs exposed to PNF extracts showed significantly higher cell viability, faster wound closure, and superior odontogenic differentiation. This was apparent through Alizarin Red staining of calcified nodules, elevated alkaline phosphatase activity, and increased expression of osteo/odontogenic markers. Moreover, HPLC analysis revealed a higher release of TEDGMA, UDMA, and BISGMA from AEF.

## Conclusions

PNF showed better cytocompatibility and enhancement of odontogenic differentiation than AEF.

## Introduction

In the ever-evolving field of dentistry, developing and applying advanced materials play a crucial role in enhancing treatment outcomes. Among these materials, flowable composites have evolved as a significant innovation. Flowable resin composites (FRC) are tooth-colored restorative materials that contain a lower filler particle content, making them suitable for specific applications such as small cavities, areas with difficult access, non-carious cervical lesions, or as a liner under other restorative materials [1,2]. Small-gauge dispensers are used to apply them, streamlining the placement procedure and improving flow and flexibility. However, FRC has reduced wear resistance, strength, and color stability compared to its bulk counterparts [3].

FRC consists of resin matrix and inorganic fillers. The resin matrix consists typically of methacrylate monomers such as bisphenol A-glycidyl methacrylate (Bis-GMA), urethane dimethacrylate (UDMA), trimethylene glycol dimethacrylate (TEGDMA), hydroxyethyl methacrylate (HEMA), or decanediol dimethacrylate (D3MA) [4]. The resin matrix principally contributes to FRC's mechanical properties, adhesion, aesthetics, biocompatibility, and handling characteristics. During polymerization, some chemical compounds such as (co)monomers, additives, and non-polymerization products are released, which can potentially cause a cytotoxic effect on the related tissue, most notably the tooth pulp [5,6].

The inorganic fillers are usually made of glass, quartz, or ceramic materials. The filler particles reinforce the composite, improve its mechanical properties, and determine its shade and translucency [2,7]. Recent advancements in FRC resulted in the introduction of improved formulations with higher filler content. One notable example is a highly filled flowable resin composite, boasting a filler content of 69 percentage by weight (wt%), with claims of greater strength, improved wear resistance, and superior gloss retention. The use of this material has been recently reported in case reports, particularly in conjunction with the "injectable resin composite technique" [2].

Flowable nano-hybrid composites are a type of dental composite material that combines the qualities of nano-filled and micro-filled composites. They contain nanoscale filler particles ranging in size from 1nm to 100nm that contribute to enhanced polishability and a smoother surface finish, which are both qualities that contribute to a better esthetic outcome. These nano-scale particles in the composite matrix contribute to a more uniform and densely packed structure. This uniformity reduces voids and gaps within the composite, improving mechanical strength and durability. The enhanced stability is particularly beneficial in withstanding the

daily mechanical stresses such as chewing and grinding, thus extending the lifespan of the dental restoration [8].

While these advances in dental materials offer improved outcomes and greater satisfaction in restorative dentistry, they also bring forth considerations regarding their biological interactions, particularly with dental tissues. Dental pulp stem cells (DPSCs) are emphasized in animal and human models as the primary element for the repair and rejuvenation of dental pulp [9,10]. DPSCs can be damaged through various mechanisms. Common factors that can damage or compromise the viability and function of DPSCs include physical trauma, bacterial infection due to untreated or poorly managed dental caries, and chemical irritants resulting from dental materials [11–13]. These irritants can cause damage to the cell barrier, oxidative stress, inflammation, and even cell death if they can spread through dentinal tubules into the tooth pulp [14–16]. Moreover, composite restorations might influence the odontogenic differentiation of DPSCs [17].

Besides these factors, the intrinsic physical characteristics of dental materials can also influence the well-being and performance of DPSCs. Parameters such as thermal conductivity, coefficient of expansion, and even surface roughness of these materials can influence the dental pulp microenvironment. Materials with excessive heat conductivity or a high expansion coefficient might subject dental tissues to thermal or mechanical stress, potentially affecting pulp vitality. Moreover, materials with rough surfaces may provide a habitat for bacteria, increasing the likelihood of secondary caries and infections, which could adversely affect DPSCs [18].

Another aspect to consider is the potential for leaching of chemicals from dental materials. As these materials degrade over time, they can release compounds that may be toxic to DPSCs. Factors like pH changes in the oral environment, mechanical forces from mastication, and the presence of enzymes in saliva can accelerate this degradation. The enduring consequences of such leaching on the viability and functionality of DPSCs are still a subject of ongoing research [19].

Given this context, this study aimed to assess the compatibility, stemness, and odontogenic differentiation potential of DPSCs exposed to extracts from flowable light-curing nano-hybrid dental composites. The study was guided by the hypothesis that there would be no discernible difference in the biocompatibility and odontogenic potential of DPSCs when exposed to different types of flowable composites. This research aims to bridge the knowledge gap in understanding the interaction between dental materials and vital dental tissues, particularly in the context of advanced composite technologies.

## Materials and methods

All experiments have been carried out in accordance with the guidelines of the World Medical Association Declaration of Helsinki and were approved by the research ethics committee, faculty of Dentistry, The British University in Egypt (approval FD BUE REC 22–026). A written Informed consent was obtained from all subjects involved in the study.

### Composite discs extract preparation

Table 1 lists the items utilized in this study and their composition. The two composite materials were prepared as discs (5 mm diameter and 1 mm high) in sterile conditions following the

**Table 1. Flowable composites used in this study.**

| Material name | Manufacturer | Material type | Resin matrix | Filler content (wt%) |
|---|---|---|---|---|
| **Aura easyflow (AEF)** | SDI, Australia | Flowable nano-hybrid | Bis-GMA, TEGDMA, UDMA | 56% |
| **Polofil NHT Flow (PNF)** | VOCO, Germany | Flowable nano-hybrid | Bis-GMA, TEGDMA, HEMA, UDMA | >76% |

manufacturer's instructions using a 3M Elipar DeepCure-L light-emitting diode (LED) curing light. After curing the discs, their top surfaces were finished and polished and ultra-violet sterilized for 30 min [20,21]. The extract from the composite discs was prepared following the recommendations of the International Organization for Standardization 10993–5 [22]. The extraction medium was serially diluted at 1:2 and 1:4 ratios, and the material surface area/medium volume ratio was set at 3 mm$^2$/mL. The extracts were prepared by placing each composite disc in a maintenance medium in the incubator for 72 hours [23].

## DPSCs isolation and culture

Freshly extracted sound teeth (n = 7) were obtained from consented adult patients (18–25 years old) as part of their orthodontic treatment plan at The British University's dental hospital after receiving institutional review board (IRB) approval (FD BUE REC 22–026).

DPSC isolation was performed using the enzymatic digestion method as described by several researchers [24,25]. The tooth pulp tissue was finely chopped into tiny fragments measuring approximately 2 mm in diameter. This was done in a Petri dish with phosphate buffer solution (PBS) (pH 7.4) and antibiotics. The pulp tissue was enzymatically digested using 3 mg/ml of collagenase type I (Sigma-Aldrich, MO, USA) and 4 mg/ml of dispase solution (Sigma, USA) while continuously agitated for 45 minutes at a temperature of 37˚C. The cells were grown in a 75 cm$^2$ flask using Dulbecco's Modified Eagle Medium (DMEM) (Gibco Dulbecco's Modified Eagle Medium) (Gibco, Grand Island, NY, USA) supplemented with 10% fetal bovine serum (FBS) (Gibco, Grand Island, NY, USA) and antibiotic-antimycotic mixture (Lonza, USA) consisting of 1% penicillin G sodium (10,000 IU), streptomycin (10 mg), and amphotericin B (25 µg). The flask was placed in an environment with 5% $CO_2$ and incubated at 37˚C. The culture medium was replaced daily. Cells used in the subsequent experiments were obtained from passages 3 to 5 from 2 individuals to provide a biological duplicate.

## DPSCs characterization

Characterization of the DPSCs was achieved by the following:

**Inverted light microscope.** Cell morphology was examined using an inverted microscope (TCM 400, Olympus, Japan).

**Flow cytometry.** Immunophenotypic identification of DPSCs was performed using flow cytometry analysis with fluorescent-labeled monoclonal antibodies (mABs) each labeled with specific fluorochromes such as fluorescein isothiocyanate (FITC) and phycoerythrin (PE). The mABs used against CD34/PE, CD45/FITC, CD73/PE, CD90/PE, and CD105/FITC (Biosciences, CA, USA). Briefly, 100 µL of cell suspension was mixed with 10 µL of the mAB and incubated for 30 minutes in the dark at 37˚C. The cells were then washed using PBS containing 2% bovine serum albumin. The cells were resuspended in BPS and analyzed using a Facial Action Coding System (FACS) Caliber Flow Cytometer (BD Biosciences) [26,27].

**Multilineage differentiation.** The multilineage, osteogenic, adipogenic, and chondrogenic differentiation ability of DPSCs was analyzed using the multilineage differentiation kit according to the manufacturer's instructions (Identification kit, R&D Systems, Minneapolis, MN, USA). Briefly, cells were seeded in a 24-well plate using a standard complete culture medium. After reaching 70% confluence, the culture medium was aspirated and replaced by an induction medium for adipogenesis, osteogenesis, and chondrogenesis. The induction medium was changed twice per week for three consequent weeks. At the end of the induction period, adipogenesis was determined by Oil-Red staining (Sigma, USA) detecting oil droplets, osteogenesis was determined using Alizarin Red-S (ARS) staining (Sigma, USA) detecting mineralized nodules, and chondrogenesis was determined using Alcian Blue stain (Sigma,

USA) detecting sulfated proteoglycans. An inverted phase contrast microscope (TCM 400, Olympus, Japan) was used for examination following staining procedures [28].

## Cell viability assay

To determine the effect of composite extracts on DPSCs viability, 3-[4,5-dimethylthiazol-2-yl]-2,5 diphenyl tetrazolium bromide (MTT) assay was employed. Cells were seeded (1 x $10^3$ cells) in a 96-well plate with a standard culture medium. After 24 hours, the culture medium was discarded, and the medium with the composite extract of AEF and PNF was added. DPSCs viability was checked after 1, 3, and 7 days of culturing with the extracts. At the designated time points, 100 µL of MTT solution (0.5mg/ml) (Sigma, USA) was added to each well and incubated for 4 hours. Then, the supernatant was aspirated and replaced with dimethyl sulfoxide (DMSO) (100 µl/well). The violet color was measured using a microplate reader at 570 nm [23,29]. Cell viability percentage was then calculated against control according to the formula by Dahake et al., 2020 [25]. All samples were performed in triplicate, and the assay was repeated three times. Materials were classified as mildly, moderately, or highly cytotoxic if cytotoxicity values were less than 30%, between 30% and 60%, or more than 60%, respectively.

## High-performance liquid chromatography analysis

A detailed experimental procedure was executed to assess the release of monomers from the composite materials. Five discs of each composite were precisely fabricated and then submerged in 0.5 ml of serum-free culture medium (Life Technologies, Paisley, UK) for 72 hours. These discs were incubated at 37˚C and 5–10% $CO_2$.

   Following the incubation, the composite discs were removed from the medium, and an equal volume of high-grade acetonitrile, compatible with high-performance liquid chromatography (HPLC), was added. This step was critical to aid in the extraction of monomers. The mixture underwent vigorous vortexing for thorough mixing, followed by centrifugation at 100 rpm for 5 minutes, ensuring an effective separation of the compounds from the solution.

   The HPLC system was employed for the analysis, focusing on the monomers TEGDMA, Bis-GMA, and UDMA. The chromatographic analysis involved injecting 100 µl of the sample, repeated three times for each sample, into the HPLC system.

   The chromatographic mobile phase comprised 65% acetonitrile (Sigma-Aldrich, gradient grade) and 35% ultrapure water, processed through a Direct-Q 3 UV system (Millipore). For the 12-minute run, the flow rate was kept constant at 1 mL/min, and the measurements were performed at a wavelength of 205 nm.

   Data analysis was conducted using liquid chromatography (LC) Solution software (Shimadzu, Kyoto, Japan), adhering to the methodology outlined by Pelka et al. [30]. Standard calibration solutions were prepared by dissolving 10 mg of each monomer in 4 mL of an acetonitrile/water mixture (1:1 ratio). These solutions were then introduced into the HPLC system to generate standard spectra.

   The peak values and retention periods for every monomer were carefully noted throughout the investigation. The monomers' concentrations in µmol/L were then determined using these measurements. This computation, which provided a thorough quantification and characterization of the monomers released from the nano-hybrid dental composite materials, was based on the area under the curve of the peaks of the standard solutions. The monomers standards utilized in the HPLC test are shown in Table 2.

**Table 2. The standards used in the study.**

| Monomers | Name | Chemical Formula | Molecular Weight |
|---|---|---|---|
| Bis-GMA | Bisphenol A glycidyl methacrylate | $C_{29}H_{36}O_8$ | 513.00 |
| TEGDMA | Triethylene glycol dimethacrylate | $C_{14}H_{22}O_6$ | 286.32 |
| UDMA | Urethane dimethacrylate | $C_{23}H_{38}N2O_8$ | 470.56 |

## Cell migration assay

Cells were cultured in a 6-well plate till reaching 100% confluence. Cells were divided into three groups: the control group (cells were cultured in regular culture medium), the AEF extract group, and the PNF extract group. A scratch was induced in each well using a sterile P-200 pipette tip through the monolayer. The cells were then washed carefully with PBS to remove debris. Images were captured initially after the scratch was induced (day 0) and after 1, 2, and 3 days. The wound size was measured from 3 images from the same frame at each time point, and the wound closure percentage was calculated [31,32].

## Dentinogenic differentiation assay

The DPSCs were seeded in 6-well plates (1 x $10^5$) and cultured in a standard culture medium till reaching 70% confluence. To induce odontogenesis, the culture medium was replaced with induction osteogenic medium (OM), which consisted of α-MEM medium (Gibco, Grand Island, NY, USA) supplemented with 100 nM dexamethasone, 200 μM ascorbic acid-2 phosphate, and 10 mM β-glycerophosphate (Sigma Aldrich, Steinheim, Germany). Cells were divided into four groups: cells cultured in OM with the AEF extract, cells cultured in OM with the PNF extract, cells cultured in standard culture medium served as negative control, and cells cultured in OM only served as positive control. The odontogenic differentiation ability of the cells was assessed after 14 days using Alizarin Red (AR) staining and quantification and alkaline phosphatase (ALP) activity assay.

## Alizarin red assay

After odontogenesis, the medium was removed, and the cells were fixed with a 10% formaldehyde solution for 15 minutes at room temperature. The wells were then washed twice with PBS to remove any non-adherent cells, and then the monolayer was stained using a 20% ARS solution (pH 4.2) for 20 minutes in the dark at room temperature. The cells were then washed four times with PBS to remove excess stain, and the formed mineralized nodules were imaged by an inverted microscope. The ARS was then solubilized using a 10% glacial acetic acid, and the produced color was measured by a spectrophotometer (Thermo Fischer Scientific, USA) at 405 nm. All samples were performed in triplicate, and the experiment was repeated three times [33].

## Alkaline phosphatase assay

Alkaline phosphatase (ALP) enzyme activity was measured by measuring the rate of conversion of the colorless para-nitrophenolate phosphate (p-NPP) into the yellow-colored para-nitrophenolate (p-NP). The monolayers of the cells were rinsed with PBS and then rinsed twice with alkaline phosphatase buffer (ALPB). Subsequently, 1 ml of the ALPB was added to each well and mixed with an equal volume of p-NPP solution (Sigma, Germany). Immediately, an aliquot of 50 μL was taken and mixed with an equal volume of 1M sodium hydroxide solution in a 96-well plate denoting time zero. The previous sampling step was repeated every

**Table 3. Primer sequences for genes used in this experiment.**

| Gene | Forward Sequence | Reverse Sequence |
|------|------------------|------------------|
| OPG | CTAATTCAGAAAGGAAATGC | GCTGAGTGTTCTGGTGGACA |
| RUNX2 | GTTATGAAAAACCAAGTAGCCAGGT | GTAATCTGACTCTGTCCTTGTGGAT |
| DSPP | TCACAAGGGAGAAGGGAATG | TGCCATTTGCTGTGATGTTT |
| β-actin | TCCGTCGCCGGTCCACACCC | TCACCAACTGGGACGATATG |

minute for 10 minutes per well, and the p-NP yellow color was measured using a spectrophotometer at 405 nm. The rate of p-NP accumulation was plotted against time for each group, and the slope, indicating the reaction rate, for each reaction was determined [34].

### Real-time Quantitative Reverse-transcriptase Polymerase Chain Reaction (RT-qPCR)

The messenger ribonucleic acid (mRNA) expression levels of osteogenic markers osteoprotegerin (*OPG*), RUNX Family Transcription Factor 2 (*RUNX2*), and the odontogenic marker dentin sialophosphoprotein (*DSPP*) were detected by RT-PCR. Total RNA extraction was performed using Trisol (Zymo Research Corp. Irvine, CA, USA). Complementary deoxyribonucleic acid (cDNA) synthesis was performed using the miScript Reverse Transcription Kit (QIAGEN), and qPCR was performed using Maxima Syber Green qPCR Master Mix (Thermofisher Scientific, Lithuania) according to the manufacturer instructions. mRNA expression levels were normalized to *β-actin* [35]. Gene expression was determined using the 2−ΔΔCt formula [36]. A list of the primer sequences used in this study is shown in Table 3.

### Data analysis

Using Statistical Package for the Social Sciences (SPSS) for Windows, statistical analysis was carried out for each experiment; a p-value of less than 0.05 was deemed significant. Every experiment was carried out three times independently and all groups were performed in triplicate. Standard deviations and means were used to express the data. MTT assay was analyzed statistically by unpaired t-test while all other assays were analyzed using ANOVA test followed by the Tukey post-hoc analysis. $p$-value $< 0.05$ was considered significant.

## Results

### Isolation and characterization of DPSCs

Isolated DPSCs displayed a spindle shape and fibroblast-like morphology (Fig 1A). The surface antigen expression demonstrated that cells maintained the characteristic surface markers of mesenchymal stem cells (MSCs), with positive CD73, CD90, and CD105 expression and negative for the hematopoietic origin markers CD45 and CD34 (Fig 1B).

DPSCs also showed a trilineage differentiation capability into adipogenic, chondrogenic, and osteogenic precursors. Osteogenesis of DPSCs was assessed using an ARS stain. The mineral deposits were detected as bright orange-red colored deposits forming a dense mineralized plexus. (Fig 2A). Adipogenesis was evaluated with Oil Red O dye. The staining revealed scattered lipid droplets. (Fig 2B). Chondrogenesis was assessed using Alcian Blue dye. The staining showed blue-colored extracellular secretions of sulfated proteoglycans (Fig 2C).

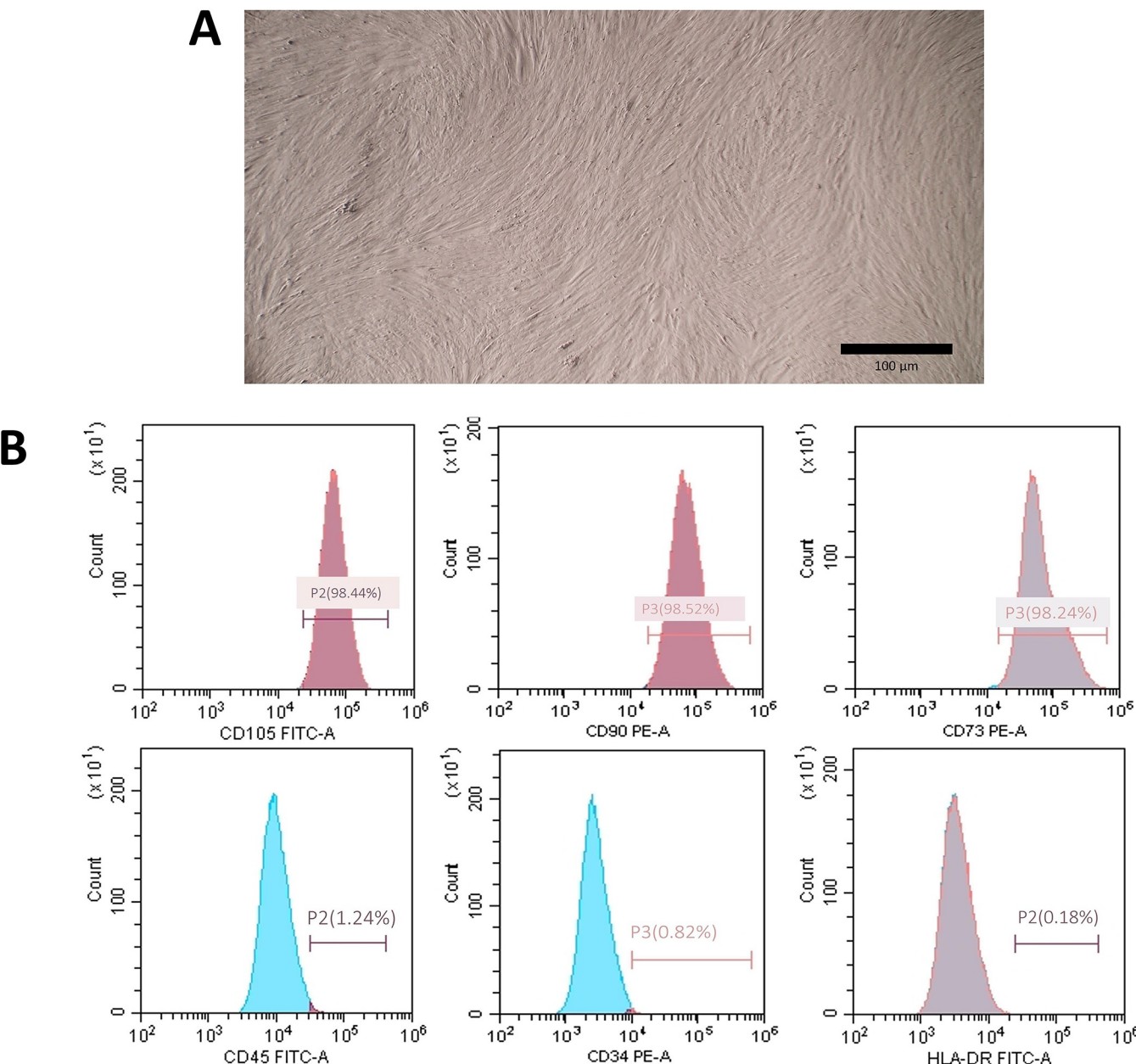

**Fig 1. Characterization of DPSCs.** A- DPSCs with fibroblast-like morphology. B- Flowcytomettry analysis of surface antigens CD105, CD90, CD73, CD45, CD34, and HLA.

### Cell viability assay

The viability of DPSCs post-exposure to PNF and AEF composite resin extracts on days 1, 3, and 7 was determined using the MTT assay (Fig 3).

**Dilution 1:1.** On Day 1, AEF and PNF groups showed similar cell viability, around 100%. On Day 3, AEF showed a slight decrease in viability, while PNF showed a more significant reduction, but no statistical significance was indicated between them. By Day 7, AEF's viability further decreased to approximately 60% and PNF's to around 50%. There is a statistically significant difference between AEF and PNF on Day 7, with AEF showing higher viability.

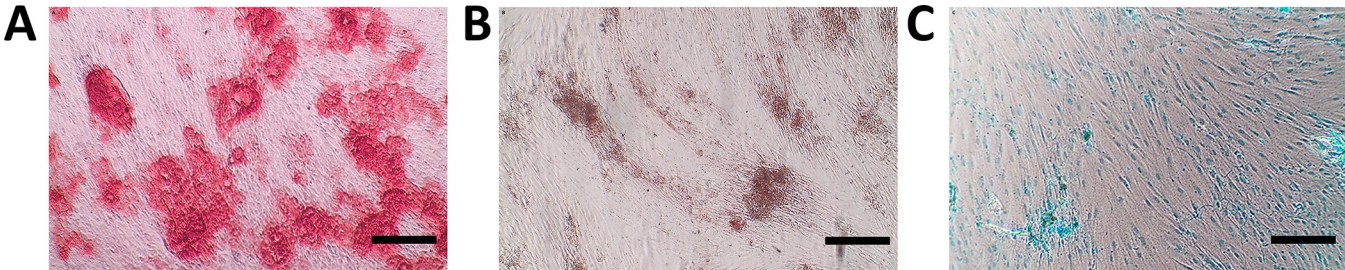

**Fig 2. In vitro differentiation assays.** A: Photomicrographs of ARS staining of DPSCs showing red calcium phosphate deposit staining confirm osteogenic differentiation. B: Photomicrographs of Oil Red O staining showing brownish-red oil droplets confirm adipogenic differentiation. C: Photomicrographs of Alcian Blue staining showing blue staining of sulfated proteoglycans confirm chondrogenic differentiation.

**Dilution 1:2.** On Day 1, the viability of both AEF and PNF is around 100%, with no significant difference. On Day 3, AEF's viability was slightly reduced, while PNF significantly reduced to around 75%. By Day 7, AEF's viability decreased to around 80%, and PNF's to about 50%.

**Dilution 1:4.** On Day 1, AEF and PNF had close to 100% cell viability. On Day 3, both groups showed reduced cell viability, with AEF at about 90% and PNF at around 80%. On Day 7, AEF's viability was roughly 80%, and PNF's was about 70%.

## High-performance liquid chromatography (HPLC) analysis

Throughout the investigation, monomers were consistently seen to be released from every composite specimen. As shown in Fig 4 and Table 4, the study showed a statistically significant difference (p < 0.05) between the composite resins, the types of monomers released, and the measurement days.

**TEGDMA.** Both AEF and PNF show a steep decline in the concentration of TEGDMA from Day 0 to Day 2. After Day 2, the concentration of TEGDMA for both substances stabilizes, with AEF showing a slight decrease and PNF remaining almost constant at around 2 μg/ml.

**BISGMA.** AEF starts at approximately 0.8 μg/ml and shows a slight decrease by Day 8. PNF begins at a lower concentration (around 0.6 μg/ml) and maintains a relatively steady level throughout the period, ending slightly below 0.6 μg/ml. For both substances, the decrease in BISGMA concentration is less steep than in TEGDMA.

**UDMA.** AEF shows a more pronounced decrease from approximately 3.5 μg/ml to about 2.5 μg/ml. PNF starts at a lower concentration than AEF and decreases to just above 1 μg/ml by Day 8. Both composites indicate a steady decline over the 8 days.

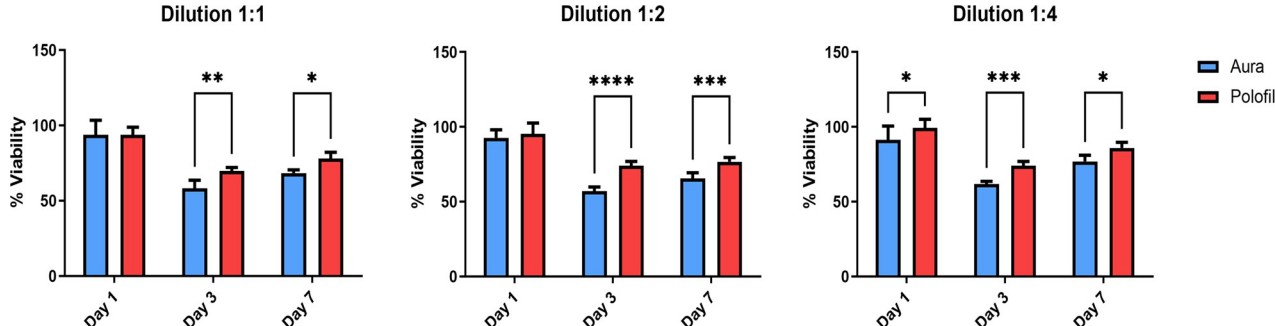

**Fig 3. Cell viability assay.** Mean, standard deviation, and comparative statistics of cell viability percentage of DPSCs after exposure to flowable composite discs extracts at Day 1, 3, and 7. *p < 0.05, **p < 0.01, ***p < 0.001, **** p < 0.0001 indicate significance.

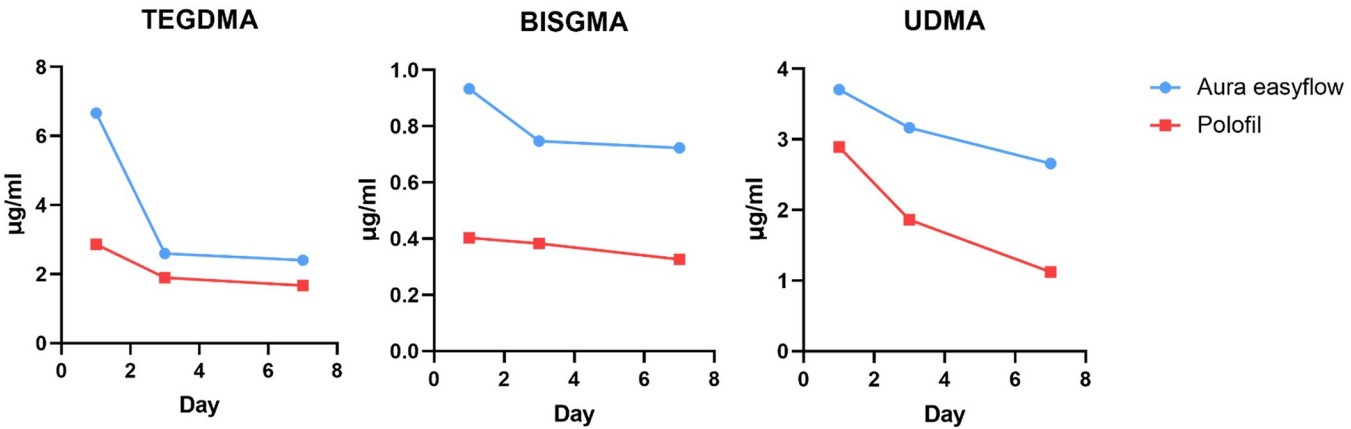

**Fig 4. Mean monomer release (ug/ml) of TEGDMA, BISGMA, and UDMA from nano-hybrid flowable composites at days 1, 3, and 7.**

## Cell migration assay

Cell migration was significantly higher in the control group than in both composite groups on all three days. Comparing both flowable composites, PNF showed significantly faster wound closure than the AEF group on all durations (p = 0.0008, p = 0.0005, p = 0.0012 in days 1, 2, and 3, respectively) (Figs 5 and 6).

## Alizarin Red (AR) staining

After cell differentiation, the calcified nodules were detected using ARS stain for all groups. The ARS staining showed that both AEF and PNF treatments lead to the formation of calcium deposits, with PNF being more effective than AEF based on the quantification of the staining. The PNF group showed significantly more calcified nodules than the AEF and the negative control group (p <0.0001) and were comparable to the osteogenic group (Fig 7).

## ALP activity assay

The PNF group had a significant advantage over the AEF group regarding ALP activity (p = 0.0035). The results for both materials were non-significant compared to the positive control group (p > 0.05) and showed significantly higher ALP activity than the control group (p<0.0001) (Fig 8).

## RT-PCR

PNF and AEF extracts exhibited elevated *OPG* gene expression compared to the control group (p = 0.0089, p = 0.9961), but their expression levels were lower than those in the osteogenic

**Table 4. HPLC analysis.**

| Monomer | Day 1 | | Day 3 | | Day 7 | |
|---|---|---|---|---|---|---|
| | **AEF** | **PNF** | **AEF** | **PNF** | **AEF** | **PNF** |
| **TEGDMA** | 6.66$^A$±0.48 | 2.86$^B$±0.07 | 2.59$^A$±0.16 | 1.89$^B$±0.07 | 2.4$^A$±0.17 | 1.67$^B$±0.08 |
| **BISGMA** | 0.93A±0.02 | 0.4$^B$±0 | 0.75$^A$±0.02 | 0.38$^B$±0 | 0.72$^A$±0.01 | 0.33$^B$±0 |
| **UDMA** | 3.7$^A$±0.03 | 2.89$^B$±0.05 | 3.16$^A$±0.04 | 1.86$^B$±0.05 | 2.65$^A$±0.12 | 1.12$^B$±0.03 |

Mean concentrations of the released monomers (ug/ml) after 1, 3, and 7 days. Different superscripts indicate significance at p<0.05.

| Control | AEF | PNF |
|---|---|---|

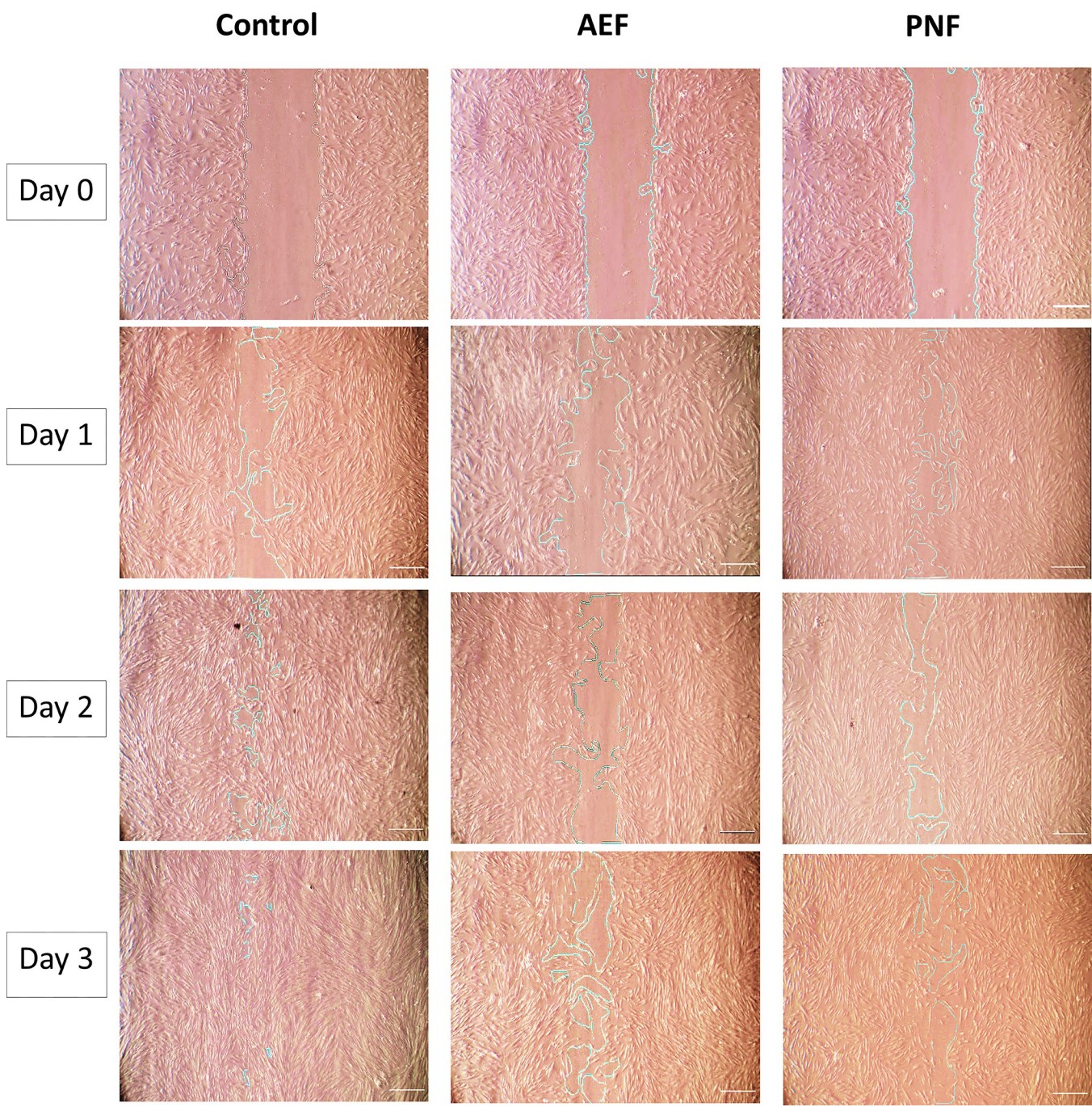

**Fig 5. Cell migration assay.** Photomicrographs showing wound closure at the initiation of wound (day 0) and after 1, 2, and 3 days.

group (p = 0.8968, p = 0.0049). Notably, the PNF group demonstrated significantly higher gene expression than the AEF group (p = 0.0118). Regarding RUNX2 gene expression, the PNF group displayed significantly greater levels than the AEF group (p = 0.0008). Similarly, regarding *DSPP* gene expression, the PNF group exhibited significantly higher levels than the AEF group (p = 0.0209) (Fig 9).

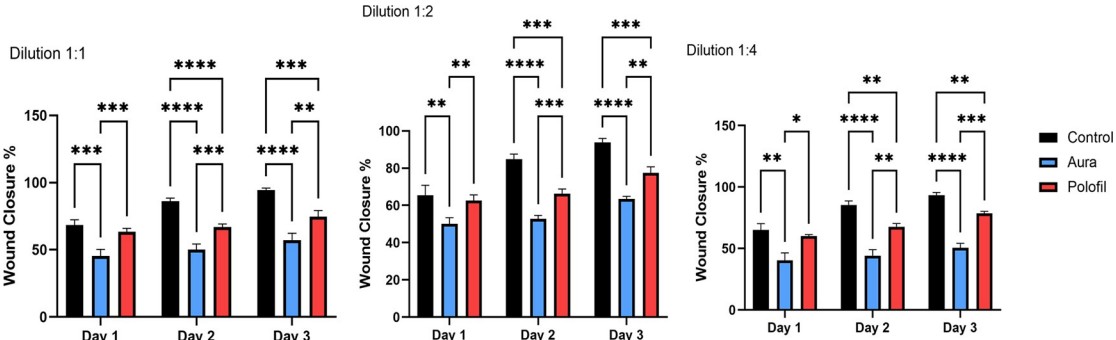

**Fig 6. Cell migration statistics** Bar chart showing mean, standard deviation, and comparative statistical results of wound closure percentage on days 1, 2, and 3 of different dilutions of eluates, (a) Undiluted, (b) Dilution 1:2, and (c) Dilution 1:4 $^*p < 0.05$, $^{**}p < 0.01$, $^{***}p < 0.001$, $^{****} p < 0.0001$ indicates significance.

## Discussion

This study investigated the response of DPSCs to extracts of recently introduced nano-hybrid FRC, PNF, and AEF. Both materials incorporate the nano-hybrid technology, which combines nano- and micro-sized filler particles to improve their mechanical and wear resistance. Yet, the variations in their chemical makeup may cause different interactions with DPSCs, which we sought to clarify.

Examining the effects of dental materials on DPSCs is crucial, given the pivotal role these cells play in maintaining the health and function of dental pulp. Using two key assays, this study evaluated the biocompatibility of flowable resin composite (FRC) extracts with DPSCs. The MTT assay provided a quantitative analysis of cell viability, while the wound healing closure assay offered insights into cell proliferation and migration. These in vitro biological assays

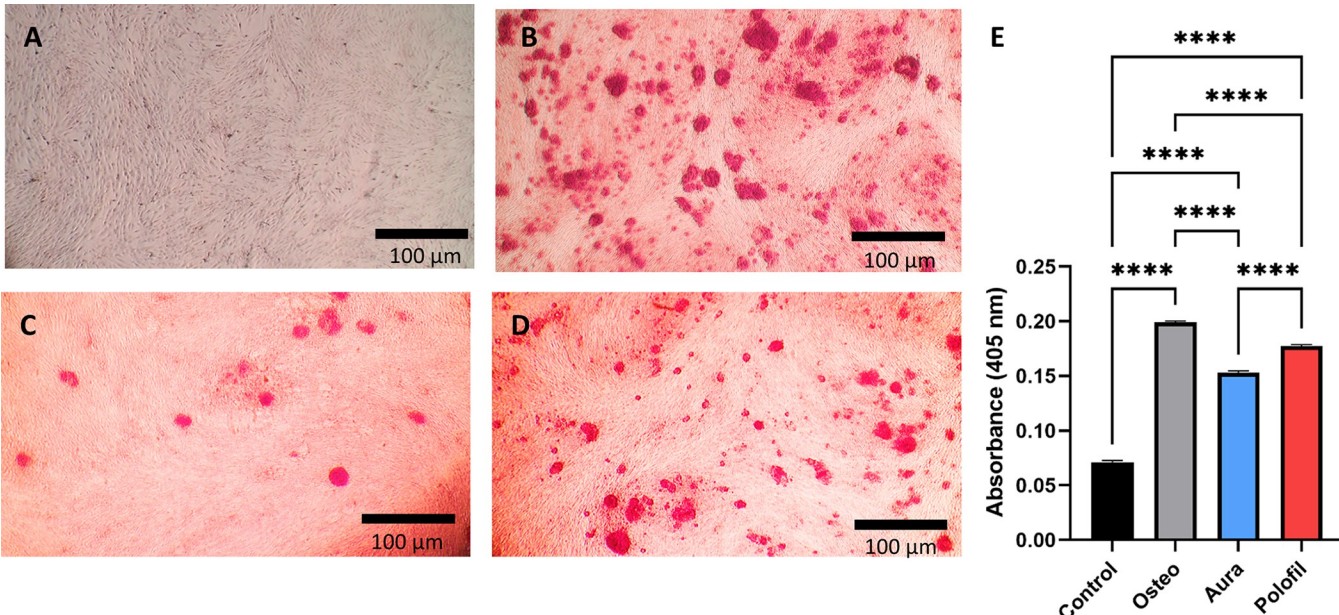

**Fig 7. Alizarin red staining.** Photomicrographs of ARS staining for the control group (A), Osteogenic group (B), AEF (C), and PNF (D) (x40 original magnification). (E) Bar chart of mean and standard deviation of the absorbance of ARS staining. $^{****} p < 0.0001$ indicates significance.

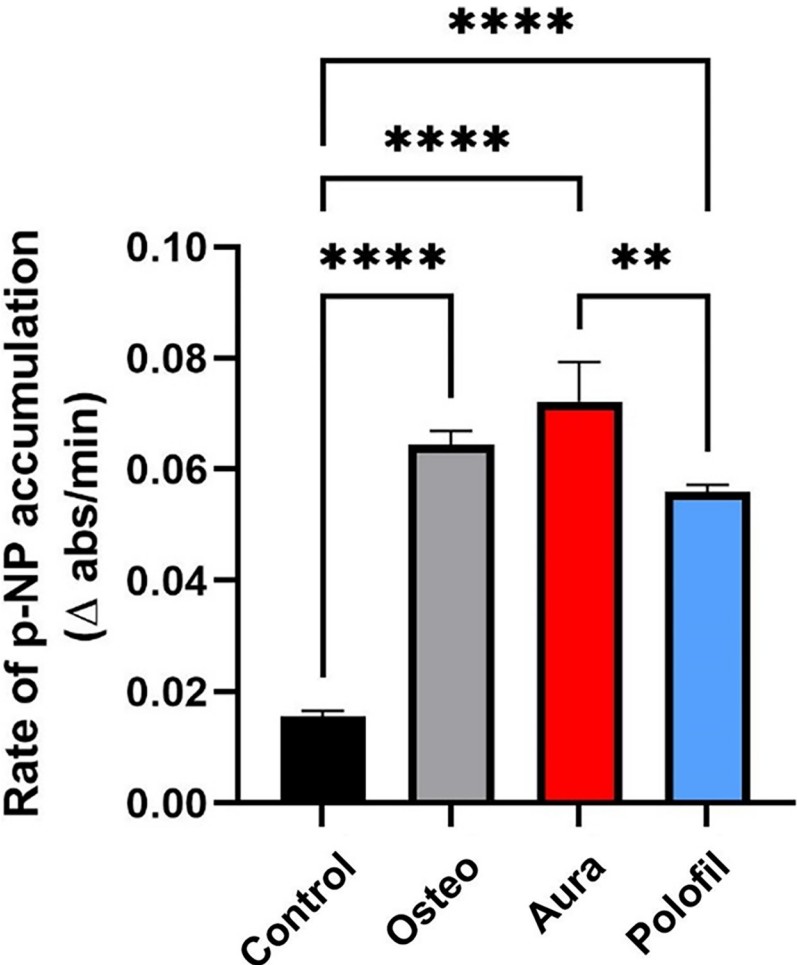

**Fig 8. ALP activity assay.** The graph represents the enzymatic activity of ALP measured as the rate of pNPP accumulation across different groups. Data are presented as mean ± standard deviation. $^{*}p < 0.05$, $^{**}p < 0.01$, $^{***}p < 0.001$, $^{****} p < 0.0001$ indicates significance.

provide several advantages, including a consistent and straightforward research methodology, the ability to test multiple materials under controlled conditions simultaneously, the requirement of only minimal quantities of the materials being tested, and a significantly shorter duration for testing compared to in vivo experiments [37].

All test groups exhibited mild cytotoxic effects DPSCs with cell viability exceeding 70% on the first day. However, distinct differences emerged on days 3 and 7. During this period, PNF continued to show only mild cytotoxicity, while AEF exhibited moderate cytotoxicity, with cell viability ranging between 40–70%. This indicates a higher proportion of non-viable or damaged cells in the AEF group. The varying cytotoxic responses of these FRCs can be attributed to differences in their chemical compositions and structures.

Despite having a comparable resin matrix, PNF and AEF are both nano-hybrid FRCs; nonetheless, their filler contents are very different. Filler content has a direct bearing on FRC solubility and component release rates. Research has indicated that composites with greater filler typically exhibit lower solvent absorption rates, resulting in a lower release of potentially hazardous components [38,39]. PNF stands out for having a filler content that exceeds 83% by weight, which could be a factor in its improved cytocompatibility and decreased cytotoxicity.

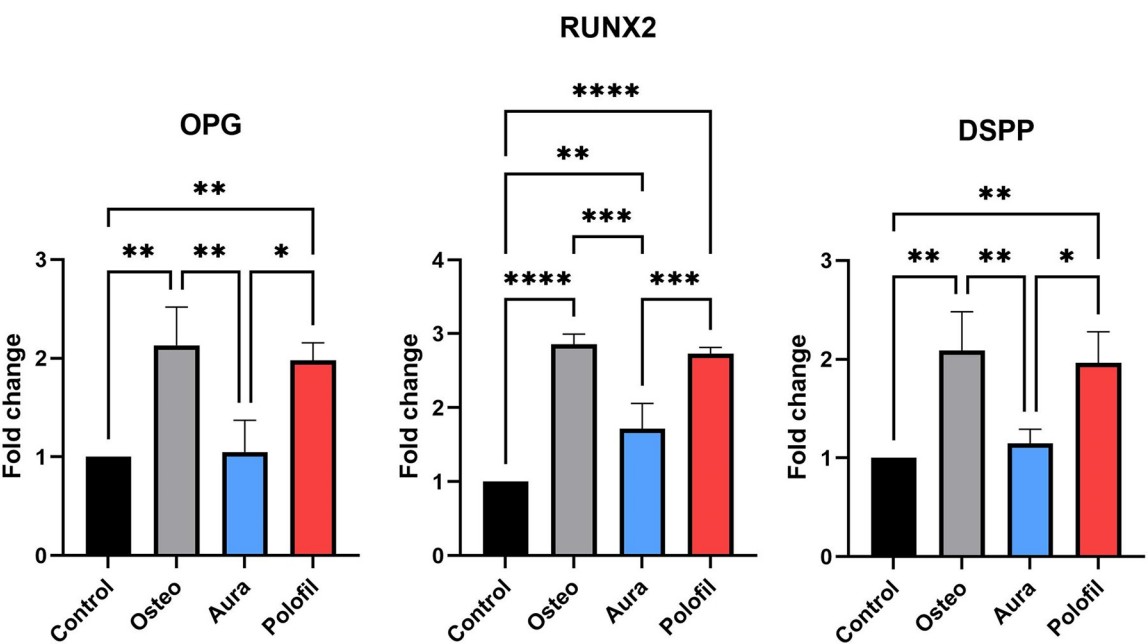

**Fig 9. RT-PCR.** Bar chart showing relative gene expression of *OPG*, *RUNX2*, and *DSPP*. Data are presented as mean and standard deviation. *p < 0.05, **p < 0.01, ***p < 0.001, **** p < 0.0001 indicates significance.

On the other hand, AEF contains 56% less filler by weight, which may be the reason for its increased cytotoxicity.

The cell migration assay outcomes reinforced the observed variances in cytotoxicity. This assay revealed that DPSCs subjected to PNF extracts exhibited a notably quicker rate of wound closure than those treated with AEF extracts. This implies that the increased filler content in PNF diminishes cytotoxicity and enhances cellular activities concerning migration and wound healing. These findings emphasize the material composition's significance in influencing dental composites' biocompatibility.

Furthermore, this study's HPLC investigation revealed a noticeably increased release of TEDGMA, UDMA, and BISGMA monomers from AEF. The molecular weights of these monomers vary; TEGDMA has the most negligible molecular weight at 286.32 g/mol, UDMA is next at 470.56 g/mol, and Bis-GMA is the heaviest at 512.60 g/mol. Interestingly, there is a relationship between these monomers' molecular weight and toxicity; studies have shown that toxicity tends to rise with molecular weight in the following order: TEGDMA < UDMA < Bis-GMA [40]. This implies that AEF's increased production of these heavier, more toxic monomers may cause its increased toxicity.

These monomers are detrimental to a variety of cell lines in earlier research. TEGDMA, in particular, is hazardous to various cell types and capable of causing mitochondrial damage. This damage to the mitochondria may result in reduced energy produced by the cell and induce apoptosis or planned cell death. Furthermore, gingival fibroblast cell proliferation has been reported to be disrupted by both TEGDMA and UDMA, primarily because of their capacity to lower glutathione levels in the cells. Evaluating the release of these monomers from dental composites is crucial since low glutathione levels can make cells more susceptible to harm and malfunction [41].

In this study, we evaluated the capacity of DPSCs to differentiate into odontogenic cells by investigating several vital markers. Using ARS staining, we performed a quantitative and

qualitative examination of calcified nodule formation, a crucial marker of odontogenic differentiation. This staining method is widely used to identify calcium-rich deposits in cell cultures, which act as a mineralization marker [28,35]. Significantly, the PNF group displayed a notably higher quantity of calcified nodules than the AEF and the negative control groups, mirroring the levels observed in the positive control group. This implies that PNF does not have an adverse effect on, and may even enhance, the mineralization potential of DPSCs, in contrast to AEF, which seemed to diminish these capabilities.

Further, ALP activity was evaluated, which is a vital marker of early odontogenic differentiation and a key player in mineralization and bone formation processes. ALP is recognized as a stemness marker in DPSCs, providing insights into their differentiation potential [42,43]. The PNF group showed a marked increase in ALP activity compared to the AEF group, indicating enhanced early-stage mineralization potential under the influence of PNF. It's important to consider that the chemical substances released from dental composites, including various monomers and by-products from the curing process, can influence ALP activity in DPSCs, potentially affecting their differentiation and mineralization pathways.

Lastly, the gene expression of odontogenic markers was meticulously analyzed using RT-qPCR. The results revealed a significant upregulation of *DSPP*, *RUNX2*, and *OPG* in the PNF group compared to the AEF group. *DSPP*, a crucial marker for odontoblastic differentiation, is vital during the post-predentin formation phase of odontogenesis [44,45]. The increased expression of *RUNX2*, a fundamental factor in odontogenic differentiation [46], and *OPG*, an essential regulator of dentin formation [47], further corroborate the superior odontogenic differentiation capabilities of DPSCs in the presence of PNF. These findings suggest that the specific composition of FRC, including their monomers and fillers, could significantly influence the signaling pathways and gene expressions critical to odontogenic differentiation. The differential release of chemicals from these composites over time may play a pivotal role in enhancing or impeding the odontogenic processes, underscoring the importance of material selection in dental restorative procedures.

The findings of this study provide significant insights into the cytocompatibility and odontogenic differentiation potential of DPSCs when exposed to flowable nano-hybrid dental composite materials. The increased cell survival, improved wound healing, and superior odontogenic differentiation reported in DPSCs treated with PNF extracts are remarkable compared to AEF. These findings are especially significant considering the growing utilization of FRC in dental treatments. The improved efficacy of PNF indicates that its composition is better suited to the biological processes of DPSCs, which are vital for the regeneration and repair of tooth tissue. This highlights the importance of meticulously choosing dental materials and considering their biological interactions to guarantee the enduring efficacy of dental restorations and therapies.

Future research should focus on elucidating the fundamental mechanics of these interactions, aiming to develop mechanical, visually attractive, and biologically compatible materials.

## Conclusions

According to the findings of this study, PNF demonstrated significantly higher cytocompatibility than AEF to DPSCs, evidenced by enhanced cell viability and migration. Moreover, PNF had a discernible and positive influence on the odontogenic differentiation potential of DPSCs.

## Supporting information

**S1 File.**
(CSV)

**S2 File.**
(XLSX)

**S3 File.**
(XLSX)

**S4 File.**
(XLSX)

**S5 File.**
(XLSX)

**S6 File.**
(XLSX)

**S1 Graphical abstract.**
(JPG)

## Author Contributions

**Conceptualization:** Dina Rady, Nassreen Albar, Waad Khayat, Mennatullah Khalil, Mohamed Shamel.

**Data curation:** Dina Rady, Mennatullah Khalil.

**Formal analysis:** Nassreen Albar, Waad Khayat, Mennatullah Khalil, Mohamed Ramadan.

**Funding acquisition:** Nassreen Albar, Waad Khayat.

**Investigation:** Dina Rady, Mennatullah Khalil.

**Methodology:** Waad Khayat, Mennatullah Khalil, Shereen Raafat, Shehabeldin Saber, Mohamed Shamel.

**Project administration:** Mohamed Shamel.

**Resources:** Dina Rady, Nassreen Albar, Shereen Raafat, Mohamed Ramadan.

**Supervision:** Mohamed Shamel.

**Validation:** Mohamed Ramadan.

**Writing – original draft:** Dina Rady, Shehabeldin Saber, Mohamed Shamel.

**Writing – review & editing:** Shehabeldin Saber, Mohamed Shamel.

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
