## [Decision Letter · Decision Letter 0]

27 Dec 2023

PONE-D-23-38885Evaluation of dental pulp stem cells response to flowable nano-hybrid dental composites: a comparative analysisPLOS ONE

Dear Dr. Saber,

Thank you for submitting your manuscript to PLOS ONE. After careful consideration, we feel that it has merit but does not fully meet PLOS ONE’s publication criteria as it currently stands. Therefore, we invite you to submit a revised version of the manuscript that addresses the points raised during the review process.

We look forward to receiving your revised manuscript.

Kind regards,

Prof. Dr Riham M. Aly

Academic Editor

PLOS ONE

Journal Requirements:

3. We note that your Data Availability Statement is currently as follows: "All relevant data are within the manuscript and its Supporting Information files."

Reviewers' comments:

Reviewer's Responses to Questions

**Comments to the Author**

1. Is the manuscript technically sound, and do the data support the conclusions?

Reviewer #1: Partly

Reviewer #2: No

2. Has the statistical analysis been performed appropriately and rigorously? 

Reviewer #1: Yes

Reviewer #2: Yes

3. Have the authors made all data underlying the findings in their manuscript fully available?

Reviewer #1: Yes

Reviewer #2: No

4. Is the manuscript presented in an intelligible fashion and written in standard English?

Reviewer #1: Yes

Reviewer #2: No

5. Review Comments to the Author

Reviewer #1: Shehabeldin M. Saber et al. submitted an interesting work about developing a nano-hybrid dental composite. The topic was to some degree of significance, and might arouse a certain impact on this field. However, there were some issues pending addressed. In conclusion, the manuscript could be reconsidered for publication after a Major Revision. Please refer to the following comments:

A. A scheme depicting the whole picture of this study could be added at the end of Introduction.

B. For the chemicals listed in Table 2, it would be helpful to also provide the molecular structures.

C. Scale bars should be supplemented in Figure 1A, Figure 2 and Figure 7.

D. What was the unit of data in Table 4? Please demonstrate in the columns.

E. Statistical analysis should be performed for Figure 4.

F. Please include some discussion about clinical and industrial translation in Discussion Section.

G. The Conclusion Section was a bit short. Some critical data could be added therein.

Reviewer #2: Thank you for the authors for addressing this important point of research and for their efforts. However, I don’t recommend publishing of this paper in the current version because of the following causes:

1) The quality of figures is too pore; the typing is not clear and the colors of the figures is not appropriate

2) The microscopic images have poor quality with no scale bars

3) The overall English and scientific writing of the paper is not professional and should be totally revised

4) Many sections in the methodology are not mentioned clearly.

6. PLOS authors have the option to publish the peer review history of their article (what does this mean?). If published, this will include your full peer review and any attached files.

Reviewer #1: No

Reviewer #2: No

---

## [Author Response · Author response to Decision Letter 0]

17 Jan 2024

Response to reviewers

The authors would like to acknowledge the insightful feedback and suggestions regarding our manuscript. We have carefully considered each of your points and have made the required revisions (marked in the manuscript).

Reviewer #1: 

A. A scheme depicting the whole picture of this study could be added at the end of Introduction.

Response: A graphical abstract is added as figure 10.

B. For the chemicals listed in Table 2, it would be helpful to also provide the molecular structures.

Response: The molecular structure is added to Table 2. 

C. Scale bars should be supplemented in Figure 1A, Figure 2 and Figure 7.

Response: scale bars added.

D. What was the unit of data in Table 4? Please demonstrate in the columns.

Response: the unit of data is µg/ml, added to Table 4 and Figure 4.

E. Statistical analysis should be performed for Figure 4.

Response: statistical analysis of table 4 is presented in table 4.

F. Please include some discussion about clinical and industrial translation in Discussion Section.

Response: added.

G. The Conclusion Section was a bit short. Some critical data could be added therein.

Response: conclusion re-written.

Reviewer #2: 

 1) The quality of figures is too poor; the typing is not clear and the colors of the figures is not appropriate

Response: replaced with new figures with better resolution.

2) The microscopic images have poor quality with no scale bars

Response: images replaced and scale bars added.

3) The overall English and scientific writing of the paper is not professional and should be totally revised

Response: We have undertaken a comprehensive revision of the manuscript. This revision was conducted with the assistance of a professional scientific editor to ensure that the language is not only grammatically correct but also conveys our research effectively and professionally.

4) Many sections in the methodology are not mentioned clearly

Response: The methodology has been updated to include more description to ensure that the methodology is transparent and reproducible.

---

## [Decision Letter · Decision Letter 1]

26 Feb 2024

PONE-D-23-38885R1Evaluation of dental pulp stem cells response to flowable nano-hybrid dental composites: a comparative analysisPLOS ONE

Dear Dr. Saber,

Thank you for submitting your manuscript to PLOS ONE. After careful consideration, we feel that it has merit but does not fully meet PLOS ONE’s publication criteria as it currently stands. Therefore, we invite you to submit a revised version of the manuscript that addresses the points raised during the review process.

*You are requested to address the reviewers new comments, especially those regarding the methodological procedures. Details **regarding the antibodies used in the flow cytometry experiments, would be valuable details for reproduction of the study. Also, more details regarding MTT and the time points are needed. Kindly, make sure that you address* 

We look forward to receiving your revised manuscript.

Kind regards,

Riham M. Aly

Academic Editor

PLOS ONE

Reviewers' comments:

Reviewer's Responses to Questions

**Comments to the Author**

1. If the authors have adequately addressed your comments raised in a previous round of review and you feel that this manuscript is now acceptable for publication, you may indicate that here to bypass the “Comments to the Author” section, enter your conflict of interest statement in the “Confidential to Editor” section, and submit your "Accept" recommendation.

Reviewer #1: All comments have been addressed

Reviewer #2: (No Response)

2. Is the manuscript technically sound, and do the data support the conclusions?

Reviewer #1: Yes

Reviewer #2: Partly

3. Has the statistical analysis been performed appropriately and rigorously? 

Reviewer #1: Yes

Reviewer #2: Yes

4. Have the authors made all data underlying the findings in their manuscript fully available?

Reviewer #1: Yes

Reviewer #2: No

5. Is the manuscript presented in an intelligible fashion and written in standard English?

Reviewer #1: Yes

Reviewer #2: No

6. Review Comments to the Author

Reviewer #1: Thank you for inviting me to re-review this paper. After going through the response file and the manuscript, I supposed that the revision would suffice.

Reviewer #2: I would like to thank the authors for their effort in conducting the experimental work of the current study. However, the English and over all scientific writing of the paper still need major modifications and Here are some comment to help you enhancing your manuscript:

1) The abstract should be divided into background, methodology, results and conclusion. Please adjust the subheadings accordingly.

2) Please state in the abstract which odontogenic markers did you evaluate?

3) Please provide a list of abbreviations

4) Please clarify what is % wt when being used for the first time?

5) The file that I received contains two versions of the manuscript, please edit.

6) The overall English of the paper looks better but it still needed to be improved, please consider academic English revision.

7) Line 8 under the subtitle of “DPSCs isolation and culture” in the methodology, there is nothing called 2mm3, please edit.

8) At the end of the same section, there is nothing called “the experiments used cells….”,

9) In the methodology, under the subtitle of “DPSCs Characterization’, please provide the model of the used inverted microscope.

10) In the flow cytometry part in the methodology, please describe whether the used antibodies were conjugated or not and use the professional way of writing as cells were stained with……..

11) In the multilineage differentiation, please add the word of “according to manufacturer’s instructions after the multilineage kit, then briefly describe the performed procedures.

12) The methodology still contains many English errors. For example, It is culture medium not culture media

13) Under the methodology of the cell viability assay, what are the different time points at which did you added the MTT?

14) Please follow the professional English terminologies and formatting while writing the methodology, for example; the sub heading under a specific heading should be shifted to some extend and so on.

15) Please provide the name and origins of the companies for the used material and supplies.

16) The resolution of figure 1 is not clear, A and B have different back grounds and even the histograms are not well- aligned and don’t have the same size.

17) Please provide a professional presentation for Fig.2

18) The resolution and background of fig.5 is not clear.

19) I can’t see the figure legends, please provide.

7. PLOS authors have the option to publish the peer review history of their article (what does this mean?). If published, this will include your full peer review and any attached files.

Reviewer #1: No

Reviewer #2: No

---

## [Author Response · Author response to Decision Letter 1]

25 Mar 2024

1. The abstract should be divided into background, methodology, results and conclusion. Please adjust the subheadings accordingly: "Thank you for your suggestion. The abstract has been revised to include subheadings for Background, Methodology, Results, and Conclusion, as requested."

2. Please state in the abstract which odontogenic markers did you evaluate?: "We appreciate your attention to detail. In the revised abstract, we have clearly stated the odontogenic markers evaluated in our study.

3. Please provide a list of abbreviations: "A list of abbreviations used throughout the manuscript has been created and added as a supplementary file."

4. Please clarify what is % wt when being used for the first time?: "We have clarified the term '% wt' at its first use in the text to indicate 'percentage by weight'.

5. The file that I received contains two versions of the manuscript, please edit. " Following PLOS ONE guidelines, we have included both a marked-up version showing the changes made and an unmarked version in the revised submission."

6. The overall English of the paper looks better but it still needed to be improved, please consider academic English revision. "We have taken your advice and have had the manuscript reviewed and revised by a native English-speaking academic in our field to improve the academic English quality."

7. Line 8 under the subtitle of “DPSCs isolation and culture” in the methodology, there is nothing called 2mm3, please edit.: "The phrase has been adjusted."

8. At the end of the same section, there is nothing called “the experiments used cells….”: "The confusing statement at the end of the 'DPSCs isolation and culture' section has been removed and replaced with a clear description of the cell preparation for experiments."

9. In the methodology, under the subtitle of “DPSCs Characterization’, please provide the model of the used inverted microscope. "The model of the inverted microscope used for DPSCs characterization is now specified".

10. In the flow cytometry part in the methodology, please describe whether the used antibodies were conjugated or not and use the professional way of writing as cells were stained with……..: "We have revised the section on flow cytometry to specify that the antibodies used were fluorescein-conjugated and have adopted the suggested professional phrasing for cell staining."

11. In the multilineage differentiation, please add the word of “according to manufacturer’s instructions after the multilineage kit, then briefly describe the performed procedures.: "Following your suggestion, we have added 'according to the manufacturer’s instructions' after mentioning the multilineage differentiation kit and provided a brief description of the procedures performed."

12. The methodology still contains many English errors. For example, It is culture medium not culture media: "We have carefully reviewed the methodology section to correct the English language errors, including the correct use of 'culture medium'."

13. Under the methodology of the cell viability assay, what are the different time points at which did you added the MTT?: "The different time points at which the MTT assay was conducted are now clearly stated in the methodology section."

14. Please follow the professional English terminologies and formatting while writing the methodology, for example; the sub heading under a specific heading should be shifted to some extend and so on.: " Thank you for your feedback regarding the formatting of the methodology section. We would like to clarify that the formatting of the entire manuscript, including the methodology section, was meticulously done according to the specific instructions and template provided by the journal. We ensured that all headings, subheadings, and text alignments were in strict adherence to these guidelines to maintain consistency and comply with the journal's requirements."

15. Please provide the name and origins of the companies for the used material and supplies. "The names and origins of the companies for all used materials and supplies are now included in the methodology section."

16. The resolution of figure 1 is not clear, A and B have different back grounds and even the histograms are not well- aligned and don’t have the same size.: "We have enhanced the resolution of Figure 1 and ensured uniform backgrounds and alignment of histograms for clarity."

17. Please provide a professional presentation for Fig.2 "Figure 2 has been professionally redesigned for clarity and improved presentation."

18. The resolution and background of fig.5 is not clear.: "The resolution and background issues in Figure 5 have been addressed to ensure clear visualization."

19. Figure Legends: " According to the journal's submission instructions: Figure captions and legends must be inserted in the text of the manuscript, immediately following the paragraph in which the figure is first cited, figure legends are embedded directly within the manuscript at the locations of the corresponding figures, rather than being listed at the end or in a separate section."

---

## [Editor Report · Decision Letter 2]

22 Apr 2024

Evaluation of dental pulp stem cells response to flowable nano-hybrid dental composites: a comparative analysis

PONE-D-23-38885R2

Dear Dr. Saber,

We’re pleased to inform you that your manuscript has been judged scientifically suitable for publication and will be formally accepted for publication once it meets all outstanding technical requirements.

Kind regards,

Dr Riham M. Aly

Academic Editor

PLOS ONE
---

## [Editor Report · Acceptance letter]

30 Apr 2024

PONE-D-23-38885R2 

PLOS ONE

Dear Dr. Saber, 

I'm pleased to inform you that your manuscript has been deemed suitable for publication in PLOS ONE. Congratulations! Your manuscript is now being handed over to our production team.

Kind regards, 

on behalf of

Dr. Riham M. Aly 

Academic Editor

PLOS ONE